# The Effectiveness of a Four-Week Digital Physiotherapy Intervention to Improve Functional Capacity and Adherence to Intervention in Patients with Long COVID-19

**DOI:** 10.3390/ijerph19159566

**Published:** 2022-08-03

**Authors:** María-José Estebanez-Pérez, José-Manuel Pastora-Bernal, Rocío Martín-Valero

**Affiliations:** Department of Physiotherapy, Faculty of Health Science, University of Malaga, 29071 Málaga, Spain; jmpastora@uma.es (J.-M.P.-B.); rovalemas@uma.es (R.M.-V.)

**Keywords:** Long COVID-19, digital physiotherapy practice, telerehabilitation, functional capacity, therapeutic adherence

## Abstract

Long COVID-19 has been defined as the condition occurring in individuals with a history of probable or confirmed SARS-CoV-2 infection, with related symptoms lasting at least 2 months and not explainable by an alternative diagnosis. The practice of digital physiotherapy presents itself as a promising complementary treatment method to standard physiotherapy, playing a key role in the recovery of function in subjects who have passed the disease and who maintain some symptomatology over time. The aims of this research are to explore the effect of a digital physiotherapy intervention on functional recovery in patients diagnosed with Long COVID-19 and to identify the level of adherence to the treatment carried out. A quasi-experimental pre-post study assessed initially and at the end of the 4-week intervention the functional capacity (1-min STS and SPPB) and the adherence (software) of a total of 32 participants. After the 4-week digital physiotherapy practice intervention with an individualised and customise exercise programme, a statistically significant improvement was observed (*p* < 0.05) with a small to medium effect size, high adherence rates and values above the minimal clinically important difference (MCID). We consider our intervention feasible, safe and consistent with our objectives. However, further randomised clinical trials and studies with larger samples are needed to draw extrapolable conclusions. Trial registration NCT04742946.

## 1. Introduction

Severe Acute Respiratory Syndrome-Coronavirus 2 (SARS-CoV-2) is a new coronavirus that emerged in 2019 and causes Coronavirus Disease 2019 (COVID-19) [1,2]. The disease severity has been variable, differing from asymptomatic cases to patients requiring intensive care unit (ICU) treatment. While we expect long-term symptoms in patients recovering from severe COVID-19, particularly those who have had ICU treatment, a worrying number of reports demonstrate long-term health issues after COVID-19 [3].

The 10 most frequent COVID-19 symptoms are fatigue, respiratory limitations, muscle pain, joint pain, headache, cough, chest pain, altered smell, altered taste and diarrhoea. Other common symptoms are cognitive impairment, memory loss, anxiety and sleep disturbance [2,4,5,6].

A recent meta-analysis found that 80% of infected patients persisted for a mean persistence time of more than six months with at least one sequela, indicating the need for evaluation and rehabilitation [7]. It is therefore a multiorgan disease, with a wide and heterogeneous range of sequelae [5,8,9].

In view of this, the World Health Organization (WHO) presents the first consensus definition of Long COVID-19 as the condition occurring in individuals with a history of probable or confirmed SARS-CoV-2 infection, usually 3 months after onset, with symptoms lasting at least 2 months and not explainable by an alternative diagnosis [10]. Long COVID-19 has been recognised as a public health problem, but many questions remain unsolved [10].

### 1.1. Physiotherapy in Patients with Long COVID-19

International studies and expert consensus designed roadmaps for physiotherapists in the different stages of the coronavirus both for patients in intensive care units and during their hospital stay on the ward, defining the available evidence on each modality of intervention [2,11,12,13,14]. Physiotherapy has a very important role in the functional recovery of patients with COVID-19, both at the respiratory and motor levels in their different phases of evolution [15]. For patients with a mild-to-moderate respiratory process, the short-term goal was to gradually restore physical and psychological conditions with a series of aerobic exercises to restore their pre-hospital exercise capacity [16]. In patients with a severe/critical process, the main physiotherapy interventions will be based on: patient education, aerobic exercise, strength and training exercises, secretion drainage and ventilatory techniques and remaining active when the situation of clinical stability permits [17,18].

However, the duration of persistent symptoms following COVID-19 and the causes and underlying mechanisms remain unknown. Previous studies on post-COVID-19 patients show a moderate impairment of their physical capacity, probably caused by muscle deconditioning [19,20]. In addition, the immobilisation induced by hospitalisations and/or strict isolation at home promotes a spiral of deconditioning with a massive increase in sedentary/inactive time [21].

Physiotherapy has, therefore, an essential role, not only in the acute and critical phases of the disease but also as a cornerstone in the interdisciplinary health team for the recovery of the sequelae that this disease may leave at the level of lung function and functional capacity, which should lead to an improvement in the quality of life of patients [16,17,18,22,23,24].

In 2020, the WHO published guidelines for those patients who continue to suffer from the long-term consequences of COVID-19 and provides guidance on rehabilitation [25]. Returning to activities of daily living is therefore a priority but must be done at an appropriate, safe and individualised pace within the limits of the symptoms. The exercise intensity should not be pushed because of the risk of post-exercise fatigue. A gradual increase in exercise should be based on symptoms [26]. In 2021 the Spanish Society of Family and General Practitioners (SEMG), together with the Long COVID-19 Association, developed a clinical guideline for the Long COVID-19 patient with the recommendation of physiotherapy interventions for the treatment of the symptoms [27].

### 1.2. Digital Physiotherapy Practice

Telerehabilitation or telephysiotherapy as a new area of telemedicine can be defined as the clinical application of consultation, preventive actions, diagnosis and therapy with audio–visual links and components [28,29,30,31,32,33]. The World Confederation for Physical Therapy (WCPT) and the member organisations of the International Network of Physical Therapy Regulatory Authorities (INPTRA) proposed an international definition and purpose for the practice of digital physiotherapy; however, within the literature, there are no established or recognised global standards or agreements for the definition of digital practice, sometimes also referred to as telehealth, telemedicine and telerehabilitation [34].

Studies based on digital physiotherapy practices have published the results of effectiveness, validity, non-inference and important advantages in some neurological, cognitive, musculoskeletal and respiratory disorders [31], providing an alternative approach that can better meet the needs of the patient with respiratory and pulmonary conditions, especially in terms of ease of access and the elimination of travel in all types of patients [34].

In recent years, the use of technologies has been seen as an alternative method of delivering different rehabilitation services remotely [35]. This digital practice, by physiotherapists, is presented as a complementary and promising treatment method to standard physiotherapy, providing physiotherapy support at any time and in any place, both in health care, community settings, and at home [36]. It has also been stated that the high feasibility and acceptability of this technology can be satisfactory to obtain an improvement in the functional capacity of the patients [37].

The current health framework is presented as an opportunity to implement the advantages offered by new technologies in the field of physiotherapy and rehabilitation, as well as to seek opportunities to solve the possible shortcomings of health systems, so that research on therapeutic interventions of rapid applicability in patients with Long COVID-19 are justified. The authors hypothesised that the implementation of a digital physiotherapy intervention in Long COVID-19 participants is effective to improve the functional capacity.

### 1.3. Objectives

-To explore the effectiveness and the magnitude of the effect of a digital physiotherapy intervention on functional recovery in patients diagnosed with Long COVID-19. Secondary objective:-To assess the level of therapeutic adherence to the digital physiotherapy intervention.

## 2. Methods

### 2.1. Study Design

This research is carried out by means of a quasi-experimental clinical trial in patients diagnosed with Long COVID-19. The study follows the guidelines from the Equator network precision in reporting of telehealth interventions used in clinical trials. The Template for the Intervention Description and Replication (TIDieR) (Appendix A) checklist has been added as the Appendix A [38].

To date, no studies have reported on the use of digital physiotherapy practice programs in Long COVID-19 patients so the clinical trials provided evidence for the effect size.

Specifically, one study showed minimally detectable changes MCID > 1 point of SPPB as the response to rehabilitation interventions in similar pulmonary diseases [39]. An MCID of 3 in the 1-min STS test was found in other research after an intervention of physical training [40]. Moreover, one study investigated the sustained effect of exercise therapy in patients with low physical functions, showing an effect size of 0.5 [41]. This references value could serve as the goal for digital physiotherapy interventions.

However, an online sample size calculator was used (https://www.ai-therapy.com/psychology-statistics/sample-size-calculator (accessed on 1 December 2021)) to determine minimal sample size. Included in the calculation was a one-tailed test, and we assumed a medium effect size of 0.5, a significance level of 0.05 and power of 0.8. As the first estimate of the effect size, a sample size of *n* = 27 participants was calculated. 

Subjects were selected by non-probabilistic sampling due to the characteristics of the subjects and for the convenience of the study; with various collaborations (Ronda Regional Hospital rehabilitation area, Virgen de la Victoria University Hospital, Malaga University Hospital Carlos Haya, Long COVID-19 Andalusia Association and Primary Care areas in the Province of Malaga).

The collaborators were informed about the characteristics of the study in personal interviews and the presentation of the project. Patient recruitment seeks to ensure sociodemographic diversity in relation to social origin, gender, ethnicity and education adapted to the particularities of the reference population in Andalusia and prior information in compliance with the data protection laws.

Patients were initially supervised by the researcher to conduct training and coaching sessions and ensure the correct execution of the exercises, encouraging patient adherence. Participants were instructed to perform self-training by following the exercises on videos through the personalised programme. Individual video calls were made based on the patient’s needs.

### 2.2. Patients

The study includes adults with a diagnosis of Long COVID-19, according to the clinical history provided and initial assessment made by the investigator; participants were recruited between December 2021 and June 2022.

Inclusion Criteria: Adults over 18 years of age with a diagnosis of Long COVID-19. Participants had to reside in the autonomous community of Andalusia during the research period; have computer technology with an internet connection at home (personal computer, laptop, tablet or smartphone) and the ability and knowledge to access email.

Exclusion Criteria: Cognitive ability not suitable for the use of technological tools and presence of comorbidities that would prevent the implementation of the rehabilitation programme.

### 2.3. Intervention

Participants received a personalised digital physiotherapy programme for 4 weeks, based on individual assessments. Data were collected by an evaluator in the physiotherapy department and integrated into our research databases.

The researcher selected Physiotec software (9082-5902 QUÉBEC INC., dba Physiotec, a legal person governed by the Business Corporations Act (Chapais, QC, Canada)) and its mobile application technology. This digital physiotherapy application allows health professionals to create personalised exercise programs; hold video conferences with patients and generate videos, images and parameters of each exercise, as well as send them by email and follow-up patients through the mobile application. Therefore, this has been selected as the best solution for our needs and considered to be best suited to the needs of the participants. The programme describes the exercises to be performed, the number of sets and repetitions and the criteria for progression, which was based on the clinical guidelines published for patients with COVID-19 and Long COVID-19 and described in the Introduction section [24].

The number of synchronous sessions (1-on-1) via videoconference and WhatsApp messages was determined by the initial evaluation. Interventions were limited to 1 session per day of 45–50 min maximum and always adapted to the previous evaluation and to the needs of each patient. Patients were advised to refrain from any other specific training during the intervention period. Any deviation from the adherence and practice of the digital practice programme was recorded on a daily basis, noting any adverse incidents.

Digital physiotherapy interventions could include personalised recommendations for each patient like walking, jogging or swimming added to the supervised digital interventions based on individual patient needs, starting at a low intensity and duration and increasing gradually; 20–30-min session durations were recommended, 3–5 sessions/week, although it always depended on the sensation of fatigue and/or dyspnoea that each patient presents [18]. Progressive strength training was recommended, working 1–3 muscle groups with a load of 8–12 repetitions, with 2-min training intervals. The frequency was 3–5 sessions/week for a minimum period of 4 weeks, increasing the load by 5–10%/week [18]. Secretion drainage or ventilatory techniques to re-educate the respiratory pattern, improve ventilation, mobilise the thorax and favour secretion drainage were recommended, especially in those patients with chronic pathology prior to COVID-19 or who had a reduced pulmonary capacity to cause the disease [42].

Due to the lack of detailed knowledge of sequelae, a thorough assessment of each particular case was recommended before applying digital physiotherapeutic techniques [16]. A flow diagram of the study is shown in Figure 1.

The software and the mobile application for such an intervention were provided and financed by exclusive license by the principal researcher. An example of an individualised exercise programme can be seen in Figure 2.

### 2.4. Outcome Measures

The initial assessment included a self-made clinical interview for anamnesis. The affiliation data and sociodemographic questionnaire, including age, gender, location and other sociodemographic variables, were collected.

#### 2.4.1. Main Explanatory Variable

##### Functional Capacity

Functional assessments provide important clinical information [43]. Long COVID-19 patients experience a downward spiral, leading to reduced functional abilities to perform occupational tasks. Functional assessments provide a measure of functional capacity and information on the prognosis, disease severity and degree of disability. The following tests were performed to assess the functional capacity of the patients:

Firstly, the “sit-to-stand test in 1 min” (1-min STS). The 1-min STS test is a reliable, valid and responsive test used to measure the functional exercise capacity in patients with airway impairment and elicited a physiological response comparable to the 6-min walk test (6MWT) [44]. The 1-min sit-to-stand test (1-STS) has been proposed as an alternative to the 6MWT (a widely used approach to evaluate the functional capacity) as a reliable method for individuals with various respiratory diseases [45]. To carry out this test, the standardised protocol described by Crook and collaborators was used [44].

Secondly, the short performance physical battery test (SPPB). This is a widely used and validated test battery with high internal consistency and has been previously used in COVID-19 patient research [46]. The SPPB evaluates three points: walking speed with the 4-m walk test, strength and resistance of the lower limbs by counting the time required to perform 5 squats (sit-to-stand test) and balance by standing with feet together, in tandem and semi-tandem. Each item was evaluated with a score from 0 to 4 [47]. The maximum final score is 12. An outline and protocol of the test execution is presented in Figure 3.

#### 2.4.2. Secondary Explanatory Variable

##### Adherence

Interest in patient adherence has increased in recent years, with growing literature that shows the pervasiveness of poor adherence to appropriately prescribed interventions and medications [49]. The concept of adherence to digital interventions is roughly defined as the degree to which the user followed the programme as it was designed, which can also be paraphrased as “intended use” or “use as it is designed” [50]. Adherence to the digital physiotherapy intervention was automatically recorded by Physiotec software and its mobile app, in compliance with the scheduled sessions.

### 2.5. Data Collection Procedure and Statistical Analysis

Once the study subjects were informed, data were collected for statistical analysis. The procedure for recording the different outcomes was carried out at the baseline T.0 (Pre) and 4 weeks at the end of the intervention T.1 (Post).

The data was added to the database created for this purpose and managed by the principal researcher using exportable data tables for statistical analysis. Each of the participating subjects was assigned a study participation number. The principal investigator was in charge of the anonymisation of the data, so that the data was recorded by eliminating the link with the identifiable person.

Statistical analysis was performed by a descriptive analysis of the data before the intervention, using a paired Student’s *t*-test. Kolmogorov–Smirnov with a Lilliefors significance correction normality test was used to explore homogenisation of the samples. Differences within group pre- and post-intervention were compared using a Student’s *t*-test, and the effect size was analysed using Cohen’s d.

Two measures that helped to quantify and understand the results of a hypothesis test were the effect size and the minimal clinically important difference, which complemented the significance. The effect size was the magnitude of the result, which allowed us to provide an estimate of the extent of our findings [51]. Effect sizes of Cohen´s d were classified as none (<0.2), small (≥0.2), medium (≥0.5) or large (≥0.8) [52]. The minimal clinically important difference (MCID) is defined as the smallest difference in a score in any domain or outcome of interest that patients are able to perceive as beneficial or harmful [53].

The statistical analysis was carried out at a 95% confidence level; a *p*-value of less than *p* < 0.05 was considered statistically significant in all analyses. Statistical analysis was carried out using SPSS software v22 (University of Malaga license, Malaga, Spain).

## 3. Results

A total of 32 participants completed the 4 weeks of intervention and were included in the analysis. Four participants dropped out of the study due to personal reasons unrelated to the study. No participants were excluded. This study performed a per-protocol analysis, where only information from patients who completed the study protocol was taken [54].

The majority of the participants were female (*n* = 23; 71.9%). The mean age was 45.93 years. Most (90.6%) of the participants were not hospitalised, even though they had developed Long COVID-19; of the three participants who were hospitalised during the acute phase of the infection prior to this investigation, only two of them had ICU treatment (6.3%). No participants in this study were actually in an ICU, acute phase or hospitalised. The participant characteristics are shown in Table 1.

A low percentage of participants did not present any comorbidity (4/12.50%). The presence of comorbidities is wide and varied, including, orthopaedic, respiratory, endocrine, circulatory, inflammatory, viral and multiorgan conditions, among others. A description of the comorbidities and distribution is shown in Table 2. The frequency and distribution are shown in Figure 4, with 19 participants who had two comorbidities, 5 had only one comorbidity, 3 had three comorbidities and only one participant had more than three comorbidities.

Table 3 summarises the pre- and post-intervention data of the study population. After the 4-week digital physiotherapy practice intervention with an individualised and customised exercise programme, a statistically significant improvement was observed (*p* < 0.05). The results showed that digital physiotherapy practice in Long COVID-19 patients produce significant changes in the functional capacity of the 1-min STS test and in the SPPB test, as shown in Table 3.

The results of the digital physiotherapy interventions show a medium effect size for the SPPB test and a small effect size for the 1-min STS test. All of the SPPB test parameters were assessed: balance test, gait speed test and chair stand test, with improvements in each of the records (*p* < 0.05).

An improvement of 1.21 points was obtained for the SPPB test and 3.50 points for the 1-min STS test after the intervention, which represents a good target in our study.

An optimal level of adherence is fundamental for the proper development of the digital physiotherapy intervention. The researchers set the level of adherence at a range of 12–20 sessions, over a total of 4 weeks, as described in the intervention section. Therefore, compliance with <12 sessions implied lower adherence values and a number >20 sessions implied levels above 100% adherence.

Only one of the study participants did not comply with the indicated number of sessions, a minimum of three sessions per week. In this case, the patient only achieved a level of adherence of 66.67%. The average number of sessions completed was 18.66, with a standard deviation of 3.68 during the 4 weeks of the intervention. Eleven subjects completed more than 5 sessions per week, giving an average of 21.82 sessions and a 109% level of adherence to the intervention performed.

## 4. Discussion 

Several guidelines/recommendations for the diagnosis and management of Long COVID-19 have been published [55,56,57]; nevertheless, at present, there is no consensus regarding the algorithm of investigation and scarce evidence-based interventions [58]. The guidelines have also emphasised the importance of multidisciplinary assessments, including physiotherapists, and highlighted the setting of the goals and the formulation of personalised management plans and care plans. The guidance lack details on potentially helpful rehabilitation interventions, perhaps understandably given the current paucity of supporting evidence. Furthermore, the dangers of exercise in some patients, such as those with undiagnosed acute pericarditis or myocarditis, highlight the need for a personalised approach [10,55,56,57].

The constant and rapid updating of research during the evolution of this pandemic has led us to expect subjects of study with physical deconditioning, dyspnoea secondary to exercise and muscle atrophy, as mentioned in previous research [17,18]. However, the assessment of the participants with Long COVID-19 has been more complex from the point of view of symptomatology and the presence of comorbidities. The disparity of symptomatology as shown in Table 2 has been a real challenge for an efficient physiotherapy intervention strategy.

As Long COVID-19 patients experience a downward spiral leading to reduced functional abilities, from the point of view of physiotherapy intervention, it is essential to assess the functional capacity and its impact on their daily activities. Different tests have been suggested as possible ways to measure the physical functions, and the research team decided to use the SPPB and the 1-min STS test instruments with validity in patients with similar symptomatology [40,47]. As one publication concluded, different protocols, such as the 1-min STS, and the SPPB: valid, reliable and feasible, are new, fast and practical alternatives for assessing the functional capacity [59]. In a literature review conducted in 2021 (65) on the most important measures of functional capacity in patients with COVID-19, of the 31 articles selected, 21.2% used the SPPB test, and 12.1% used the 1-min STS.

The results showed that the functional capacity outcomes improve significantly (*p* < 0.05), reducing the level of perceived exertion, improving the ability of these patients in basic physical activities such as walking or sitting, maintaining balance while standing up and getting up from a chair, showing us the clinical relevance of the intervention. Our improvement results are in line with published data in several pathologies, such as renal [60], multiple sclerosis [61], head and neck cancer patients [62] and COVID-19 patients [63].

The main reason for patient improvement following an individualised digital physiotherapy programme could be related to the improvement in the gas exchange, the stimulation of respiratory muscles and the muscle stretch and workload exercise, which could lead to improvements in cardiopulmonary and physical function, as already suggested [63].

The digital practice in physiotherapy was presented as a tool that could provide an effective response to the health problems identified [13], and the results of this research have proven to be an effective method of intervention in Long COVID-19 patients. Our results are in line with the efficacy of the digital physiotherapy interventions in several conditions [64,65,66,67,68,69,70,71,72] and increase the scientific knowledge about their use in these patients. Another advantage of digital physiotherapy home-based rehabilitation is that treatment takes place in a familiar environment where the patient feels comfortable and safe, which seems to produce beneficial results and positive outcomes for participants, especially in individuals with difficulties in accessing hospital services [55,56]. To all this, we can add the monitoring and care by a healthcare professional through synchronous sessions (one-on-one) via videoconference and messages via WhatsApp, which makes this study decisive and differential.

Since the beginning of the pandemic, digital physiotherapy practice has been suggested as a management strategy of the disease [9] and its sequelae [73]. To our knowledge, this is the first clinical trial conducted on Long COVID-19 patients using the digital physiotherapy practice.

Unlike other studies that require software implementations in specific devices, our intervention generates few obstacles, since it is available in any device that allows an internet connection, allowing access from any location and different devices. This contrasts with other studies that require a highly complex technological platform [42]. As the patient uses his own technological devices, the rapid implementation process was guaranteed. This, however, may lead to a selection bias for those patients without access to technology, which should be considered as a limitation of the study. Other limitations are related to technical problems (disconnection and device failures) and technological difficulties that may arise in connection with the use of the technology.

The greatest disadvantage of quasi-experimental studies is that randomisation is not used, limiting the study’s ability to conclude a causal association between an intervention and an outcome [38]. Nevertheless, quasi-experimental studies are often used to assess rapid responses to outbreaks and are pragmatic within its inherent design limitations [38].

Another limitation was the short intervention duration of only 4 weeks, although recent publications have shown improvements with only 7 days of intervention [66,74].

Vaccination may contribute to a reduction in the health burden population of Long COVID-19, as shown in a recently study [54]; in this research, all participants had the complete vaccination schedule, as recommended in Spain, before being enrolled in the study. However, this study selected 32 participants: 22 were infected in 2022 and 10 during 2020, maintaining a symptomatology ranging from a year and a half to 6 months. The patients included in the study had not received a continuous physiotherapy programme prior to the start of the research, although they received education programmes about their health and symptoms. During all this time, the subjects have kept up their medical visits. The vast majority of the subjects reported feeling abandoned and desperate during the first physiotherapy assessment. That contrasted with their great aptitude for recovery, good social support through the Long COVID-19 association and dedication to the objectives of the study from the very beginning.

A study conducted with Long COVID-19 patients showed a confusing illness with many, varied and often relapsing–remitting symptoms and an uncertain prognosis; a heavy sense of loss and stigma, difficulty accessing and navigating services and difficulty being taken seriously and achieving a diagnosis [75].

Given that there have been no major difficulties in the course of the intervention, and patients have not reported any adverse effects during the intervention, we found our intervention to be feasible, safe and consistent with our objectives. Our research team is carrying out an in-depth qualitative analysis with the intention of gaining further knowledge and ideas for improving care services that complement our intervention.

## 5. Conclusions

After the 4-week digital physiotherapy practice intervention with an individualised and customised exercise programme, a statistically significant improvement was observed (*p* < 0.05) in patients diagnosed with Long COVID-19 in a functional capacity, with small and medium effect sizes and high rates of adherence and values of MCID. Therapeutic exercise implemented through digital physiotherapy practice appears to provide a promising strategy for improving outcomes related to physical conditions among patients with Long COVID-19, indicating clinical benefits and adherence to the intervention. The development of this research increases the available knowledge on the use of digital physiotherapy practice in patients with Long COVID-19. However, further randomised clinical trials and studies with larger samples and control groups are needed to draw extrapolatable conclusions. All research and outcomes data are available from the main author upon request.

## Figures and Tables

**Figure 1 ijerph-19-09566-f001:**
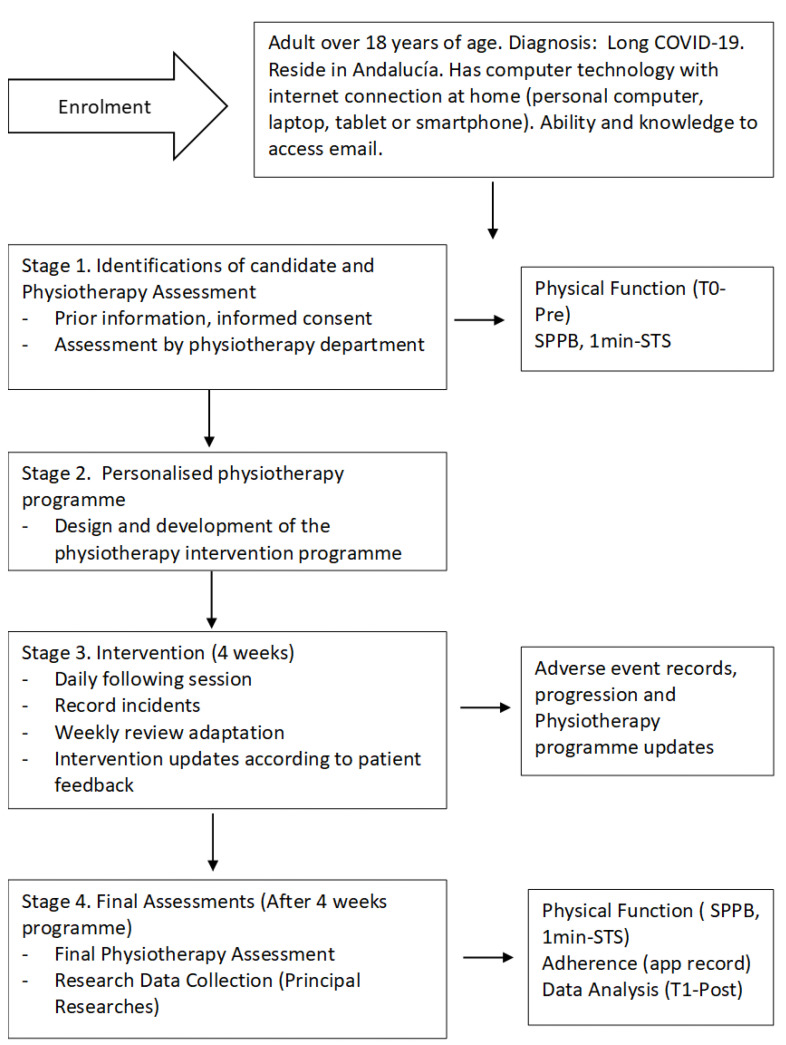
Flow diagram study design.

**Figure 2 ijerph-19-09566-f002:**
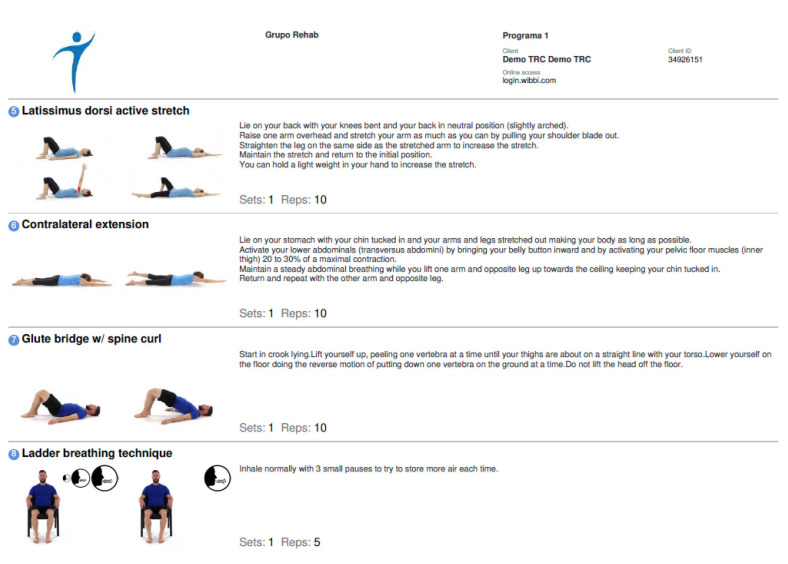
Example of an individualised exercise programme.

**Figure 3 ijerph-19-09566-f003:**
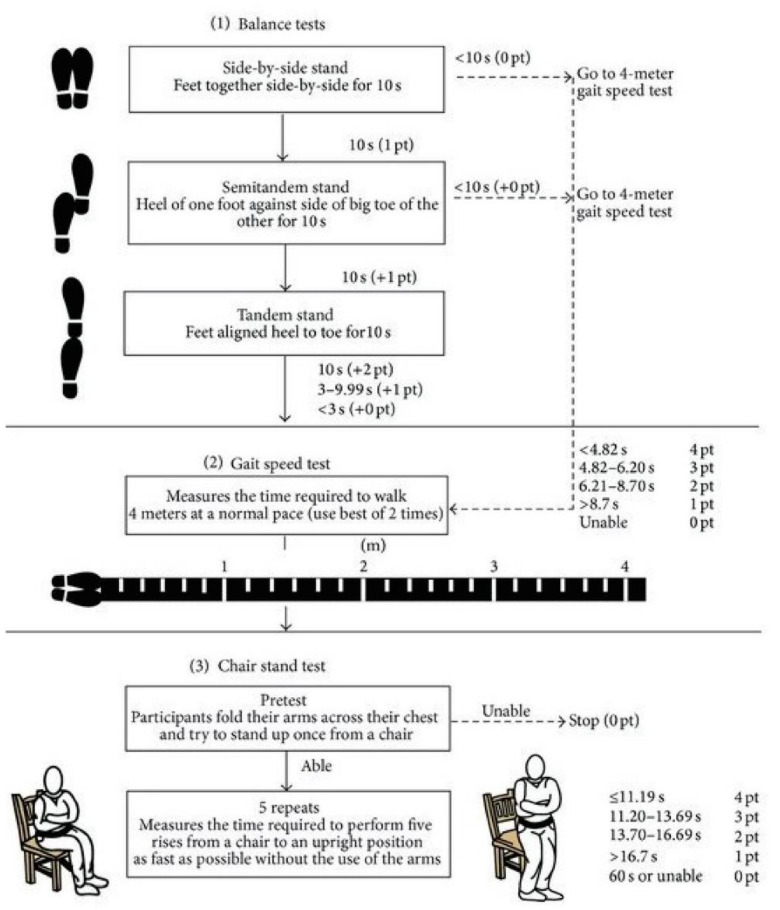
The Short Physical Performance Battery (SPPB) test. Reprinted with permission from Ref. [48]. Copyright 2021, copyright Riskowski, J.L. et al.

**Figure 4 ijerph-19-09566-f004:**
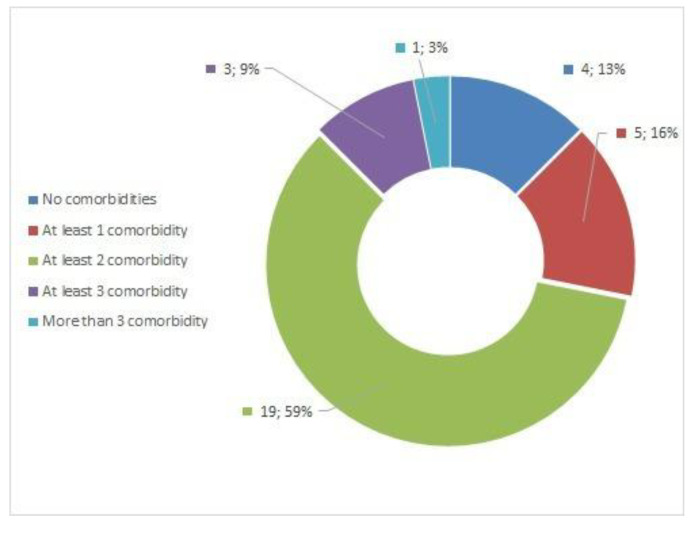
Presence of comorbidities.

**Table 1 ijerph-19-09566-t001:** Participants characteristics.

	Participants (*n* = 32)
Age (Mean/SD)	45.93 ± 10.65
Gender Woman % (*n*)	71.9% (23)
Non-Hospitalised % (*n*)	90.6% (29)
ICU % (*n*)	6.3% (2)

**Table 2 ijerph-19-09566-t002:** Comorbidities distribution.

Comorbidities Long COVID-19	*n* = 32	%
Orthopaedic Pathology (Spine Surgery, Herniated Disc, Scoliosis, Hyper lordosis, Sacral Lumbarisation, Sacroiliitis, Shoulder Tendinopathy, Osteochondritis, Osteomalacia, Osteoarthritis and Osteoarthritis)	15	46.88%
Respiratory Pathology (Asthma, Bronchial Hyperresponsiveness and Chronic Pharyngitis)	11	34.38%
Endocrine Pathology (Thyroid Pathology and Diabetes)	4	12.50%
Circulatory Pathology (Heart disease, Varicose veins and Hypertension)	3	9.38%
Inflammatory Pathology (Sarcoidosis and Pancreatitis)	2	6.25%
Depression	2	6.25%
Immunological disorders (seasonal allergies)	2	6.25%
Viral Deseases (Herpes Zoster and Mononucleosis)	2	6.25%
Headaches	2	6.25%
Myasthenia	1	3.13%
Colic	1	3.13%
Ureter Reflux	1	3.13%

**Table 3 ijerph-19-09566-t003:** Results of the 1-min STS: the 1-min STS test and SPPB: the short performance physical battery test.

	Initial (M/SD)	Final (M/SD)	Correlation	Sig.	Cohen’s d	Effect Size r	Effect Size
1-min STS	14.03/7.84	17.53/7.44	0.81	0.00	0.45	0.22	Small
SPPB	7.90/1.98	9.12/1.69	0.72	0.00	0.66	0.31	Medium
Balance test (SPPB)	3.65/0.60	3.96/0.17	0.50	0.00	0.70	0.33	Medium
Gait Speed test (SPPB)	2.87/1.09	3.34/0.90	0.72	0.00	0.46	0.22	Small
Chair Stand test (SPPB)	1.37/0.90	1.81/1.09	0.66	0.00	0.43	0.21	Small

M: Media, SD: Standard deviation, Correlation: Correlation and Sig: Bilateral signification.

## Data Availability

The researcher declared that he followed the protocols of his work centre regarding the publication of data in accordance with the provisions of Organic Law 15/1999, of 13 December, on the Protection of Personal Data (LOPD) and that the data was incorporated into a file for the purpose of carrying out this research project. Participating subjects were informed of the possibility of exercising their rights of access, rectification, cancellation and opposition of their data at the e-mail address provided by the principal investigator.

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
