# Peer review of "The Effectiveness of a Four-Week Digital Physiotherapy Intervention to Improve Functional Capacity and Adherence to Intervention in Patients with Long COVID-19"

_ijerph, 2022, doi:10.3390/ijerph19159566_

Round 1
Reviewer 1 Report
Manuscript ID: ijerph-1823030
Manuscript title: The Effectiveness of a Four-Week Digital Physiotherapy Intervention to Improve Functional Capacity and Adherence in Patients with Long COVID-19
Comments
This manuscript reports a study that explored the effectiveness and the magnitude of the effect of a digital physiotherapy intervention on functional recovery in patients with Long COVID-19. It also assessed the level of therapeutic adherence to the digital physiotherapy intervention. The background and rationale are well posed, and the study aims are objective. The Methods are appropriated to the study aims. The results are clearly reported, but requires major reviews for an accurate and transparent reporting. Discussion seems aligned with the results, but the conclusions need rephrasing due to the study design.
Major comments
1. Abstract. The conclusions should also suggest randomized clinical trials (not only larger samples) as a quasi-experimental design cannot assess the effect of the intervention.
2. Page 3, Study design. It is important to report your study following specific guidelines to increase transparency and completeness of the reporting. Please search the EQUATOR network for an appropriate guideline. I suggest checking at least the TIDier for telehealth (https://www.equator-network.org/reporting-guidelines/tidier-telehealth-precision-in-reporting-of-telehealth-interventions-used-in-clinical-trials-unique-considerations-for-the-template-for-the-intervention-description-and-replication-tidier-checkli/). I believe almost all information is already present (e.g., Figure 1 does not show the number of participants at each stage), but it is important to double-check it and organize according to the checklist.
3. Page 3, Study design. Sample size estimation is unclear. The estimated sample size of 32 was found by calculation based on the provided references? Or is it justified as being similar to previous studies? Why did you not use the minimal clinically important difference for sample size calculation? This is important because might have repercussions in Discussion section.
4. Page 8, Results. Can you provide information about how many participants were (fully) vaccinated at baseline? Vaccination status is being argued as related to Long COVID, it can be informative for future systematic reviews.
5. Page 8, Results. It is reported that 32 patients completed the 4-week trial and were included. It is also reported that 4 participants dropped out the study. Does it mean 36 patients started the 4-week trial? This has implications in summary data analysis due to how missing data was handled.
6. Page 9, Results. Given the quasi-experimental study design, I suggest rephrasing the sentence ‘The 4-week digital physiotherapy practice intervention with an individualised and customize exercise programme resulted in statistically significant improvement (p < 0,05)’ to ‘After the 4-week digital physiotherapy practice intervention with an individualised and customize exercise programme it was observed a statistically significant improvement (p < 0,05).’. It is a subtle but necessary rephrasing because you cannot claim effects of an intervention without a control group.
7. Page 9, Results. Likewise, the following sentence requires rephrasing: ‘The results confirm that digital physiotherapy practice in Long COVID-19 patient produces significant changes in the functional capacity in the 1min-STS test and in the SPPB test, as shown in table 3’.
8. Page 9, Results. The paragraph about effect size is not a result, should be moved to Statistical Analysis section.
9. Page 9, Results. Likewise, the paragraph about minimal clinically important difference should be moved to Statistical Analysis section.
10. Page 9, Results (and page 10, Figure 5). The numbers make no sense. How can a whole group split in three categories have frequencies of 100%, 109% and 66.7%? Either simple or cumulative percentage is acceptable, but in either case the maximum should be 100%. After checking the numbers, double-check if a sector plot is the best way to convey this information.
11. Page 10, Discussion. First paragraph is dispensable.
12. Page 11, Discussion. Discussion on the limitations should be expanded. There are several threats to both internal and external validities when using quasi-experimental (pre-post) designs.
13. Conclusions. The conclusions should be tone down regarding the effectiveness of the intervention given the study design. Also, it should suggest randomized clinical trials (not only larger samples) as a quasi-experimental design cannot assess the effect of the intervention.
Minor comments
1. 2.1 Study design. Did you mean ‘sample size’ rather than ‘population size’?
Reviewer 2 Report
Thank you for the opportunity to review Maria-José Estebanez-Perez and team submission titled “The Effectiveness of a Four-Week Digital Physiotherapy Intervention to Improve Functional Capacity and Adherence in Patients with Long COVID-19”. The aim of this manuscript is interesting and explore the effectiveness of a digital physiotherapy intervention on functional capacity in patients who have reported being positive to COVID-19 disease. The adherence to the digital intervention was also assessed. Overall, the manuscript is understandable even if some English corrections should be made. Some sections, like the introduction and discussion, are too extensive and do not fully address the main subject of the study i.e. telephysiotherapy intervention. Some important information is missing in the methods section especially about the digital intervention and a few points seemed to be really subjective. The sample size calculation and stats analysis need to be rechecked. Please next submission don’t forget to put the line numbers it would have helped the revision work.
Please find below more specific comments aiming to help you to improve your submission.
Title: using only ‘adherence’ in the title is confusing. We only understand that your talking about the adherence to the digital intervention in the abstract. You should be more specific.
Introduction: Overall the introduction should be rewritten. In my opinion, the introduction is too long and the authors do not focus on the main information relating to their objectives.
A few times in the manuscript, the authors only mention the existence of guidelines (WHO, WCPT, INPTRA…) for the management of patients with COVID-19 but they do not go into detail. What is missing in the introduction is: ‘why is it important to prescribe physio in patients with COVID-19 or patients who had COVID-19?” ‘what interventions are already effective for them? (physio face to face or digital physio intervention). Previous intervention found in the literature should be detailed.
Methods: for example, first sentence of the method section should be rewritten (English incorrect).
The ‘calculation of the sample size’ is an example of the subjective elements I was talking earlier. You cannot say ‘because a previous published article used 32 subjects, so I will use 32 subjects’, we need to know on what outcome you did the calculation, using a mean and SD and a specific power and alpha.
Example of elements that need to be detailed in the methods section:
- How many physio sessions were performed by participants before participating in the digital intervention?
- In the inclusion criteria did you take patients who were hospitalised in ICU?
- ‘participants received a personalized program based on clinical parameters, follow-up discharge reports and self-reported forms’ what does it mean? Be more specific.
- The researcher has selected the Physiotec… as it is considered to be best suited to the needs of the participants’ according to who? This choice of software seems really subjective.
- ‘digital physio intervention could include… walking, jogging, swimming’ I do not understand how can you swin in a digital session??? Participants were not supervised during the sessions? Was it just autonomous session?
- Where were performed the assessments? At the hospital?
- Figure 2 is nice but too small to be fully read. Few words are still in Spanish.
- Be careful using the 1min sit to stand for assessing functional capacity, as it can be used for assessing exercise capacity. (Gephine et al, 2020, MSSE).
- Another reference than Ref 47 could be mentioned for the SPPB
- As your subjects are the main before and after the intervention, you should have used a paired student test.
Results: please do not use 3 numbers after coma in table or text when mentioned numbers.
Table 1 you mentioned ‘UCI’, you mean ICU?
In my opinion Figure 4 could be removed, you already mentioned this data in the text.
Description about the effect size and MCID should be in the stats or methods section, not in the results.
Ref 55 information is misused, it was not after an intervention of physical training, but after pulmonary rehabilitation in people with chronic respiratory diseases, this should be clarified.
Regarding the level of adherence, why the authors chose 12/20 sessions? Is it a subjective opinion?
Discussion: first paragraph should be removed, it is not a thesis manuscript.
Ref 65 is incorrect it seems to be ref 64. Overall, the authors mentioned way to much references, they need to choose more appropriate/specific references.
Have you accounted for the technical issues with the digital intervention? This should be added.
The authors mentioned at the end of the discussion ‘we found our intervention to be feasible, safe and consistent with our objectives’, great, but you did not evaluate the feasibility neither the safety of your digital intervention.
Round 2
Reviewer 1 Report
Thank you for providing a response letter to my previous comments. All comments were addressed. I have no new comments.
Reviewer 2 Report
The authors addressed all my first comments. In my opinion the manuscript is can be publish in this current form. Please, pay attention to some sentences, minor spell check is required.